# A Non-Contacted Height Measurement Method in Two-Dimensional Space

**DOI:** 10.3390/s24216796

**Published:** 2024-10-23

**Authors:** Phu Nguyen Trung, Nghien Ba Nguyen, Kien Nguyen Phan, Ha Pham Van, Thao Hoang Van, Thien Nguyen, Amir Gandjbakhche

**Affiliations:** 1Faculty of Information Technology, Hanoi University of Industry, No. 298 Cau Dien, Bac Tu Liem, Hanoi 143510, Vietnam; phunt@haui.edu.vn (P.N.T.);; 2School of Electrical and Electronic Engineering, Hanoi University of Science and Technology, No. 1 Dai Co Viet, Hai Ba Trung, Hanoi 100000, Vietnam; thao.hv193119@sis.hust.edu.vn; 3*Eunice Kennedy Shriver* National Institute of Child Health and Human Development, National Institutes of Health, 49 Convent Drive, Bethesda, MD 20892-4480, USA; thien.nguyen4@nih.gov (T.N.); gandjbaa@mail.nih.gov (A.G.)

**Keywords:** non-contact, height measurement, MediaPipe

## Abstract

Height is an important health parameter employed across domains, including healthcare, aesthetics, and athletics. Numerous non-contact methods for height measurement exist; however, most are limited to assessing height in an upright posture. This study presents a non-contact approach for measuring human height in 2D space across different postures. The proposed method utilizes computer vision techniques, specifically the MediaPipe library and the YOLOv8 model, to analyze images captured with a smartphone camera. The MediaPipe library identifies and marks joint points on the human body, while the YOLOv8 model facilitates the localization of these points. To determine the actual height of an individual, a multivariate linear regression model was trained using the ratios of distances between the identified joint points. Data from 166 subjects across four distinct postures: standing upright, rotated 45 degrees, rotated 90 degrees, and kneeling were used to train and validate the model. Results indicate that the proposed method yields height measurements with a minimal error margin of approximately 1.2%. Future research will extend this approach to accommodate additional positions, such as lying down, cross-legged, and bent-legged. Furthermore, the method will be improved to account for various distances and angles of capture, thereby enhancing the flexibility and accuracy of height measurement in diverse contexts.

## 1. Introduction

It is critical to have an accurate and convenient method for measuring human height, which is an important variable in healthcare for calculating Body Mass Index (BMI) and determining various treatment-related metrics [1]. BMI enables the classification of individuals as overweight, underweight, or of ideal weight [2]. Moreover, BMI is instrumental in population-based studies due to its widespread acceptance in identifying specific body mass categories that may indicate health or social issues. Recent evidence also suggests that particular BMI ranges are associated with moderate and age-related mortality risks [3].

Height measurement is typically performed in an erect standing posture. For non-critical patients, contact methods such as table scales, standing scales, or medical measuring devices are commonly employed. However, for critically ill patients in intensive care units (ICUs), requiring them to move or assume an upright position for height measurement is often impractical [4]. Additionally, severely ill patients are frequently unconscious or incapacitated, complicating accurate height assessments. Therefore, the development of a non-contact height measurement method for critically ill patients is particularly important in the ICU setting [5,6].

Currently, height measurements for ICU patients in hospitals in Vietnam are frequently conducted by nurses; however, the nurse-to-patient ratio is often insufficient. This situation introduces significant challenges in obtaining accurate measurements. Accurate height data are crucial, as it is integral to calculating treatment parameters such as creatinine indices [7,8]. Thus, the implementation of an automatic, non-contact height measurement method represents a critical step toward ensuring the highest possible accuracy for calculating treatment parameters.

Several non-contact methods have been proposed for measuring height in special populations, such as the elderly, hospitalized individuals, bedridden patients, and those with skeletal deformities. A study conducted at Jimma University demonstrated that height estimates derived from linear body measurements, including arm span, knee height, and half-arm span, serve as useful surrogate measures [9]. However, the study was limited by a narrow age range, including only adults aged 18 to 40 years, which may not adequately represent the broader adult population, especially considering the potential decline in height in older age groups. Furthermore, Haritosh has investigated the use of facial proportions to estimate body height [10]. This method involves calculating height from facial images by extracting facial features through convolutional neural networks and predicting height using artificial neural networks. However, the average error rate in the measurement is approximately 7.3 cm, which constitutes a significant deviation in height assessment.

A common method for estimating human height from images or videos is skeletal extraction [11]. This approach utilizes computer vision and image-processing methodologies to analyze visual data. The accuracy of this method can be affected by various factors, including camera focal length, angle, and ambient-lighting conditions. To enhance the precision of height measurements, we propose a study employing MediaPipe to extract skeletal point coordinates from images capturing both a person and a reference object—a black cardboard of fixed dimensions. These coordinates, represented in a two-dimensional space as X and Y values, are used to calculate the lengths of bone segments, thus facilitating height estimation. Following the extraction of skeletal points, a machine-learning model will be employed to train the input data and estimate human height. We hypothesize that the use of a reference object will improve the accuracy of height measurement.

## 2. Materials and Methods

### 2.1. The Proposed Method

Figure 1 illustrates a diagram of the proposed height measurement method. The block diagram consists of six primary blocks. The first block serves as the input, which is an image of a person in a vertical position. The second block identifies and marks human body landmarks (skeleton points) using the OpenCV and MediaPipe libraries. The third block calculates the length of each skeleton using the MediaPipe library. This step involves calculating a centimeter-per-pixel (cm/pixel) ratio using a reference object, counting the number of pixels in each skeleton, and then calculating the skeleton length in centimeters. In the fourth block, the lengths of the skeletons are fed into a multivariate linear regression model to train the model. In the fifth block, human height is predicted using the trained model. Finally, in the sixth block, human height is obtained.

The OpenCV (Open Computer Vision) is a leading open-source library for computer vision, machine learning, and image processing. It is written in C/C++, which enables it to achieve very fast calculation speeds and allows for use in real-time applications [12]. MediaPipe is a series of cross-platform machine-learning solutions used for tasks such as face detection, face mesh, and human pose estimation [13]. It consists of three main parts, namely a framework for inference from sensory data, a set of tools for performance evaluation, and a collection of reusable inference and processing components [14]. YOLOv8 is a computer vision model for object recognition and detection developed by Ultralytics in 2016 [15]. Among different object detection algorithms, the YOLO (You Only Look Once) framework has stood out for its remarkable balance of speed and accuracy, enabling the rapid and reliable identification of objects in images. Since its inception, the YOLO family has evolved through multiple iterations, each building upon previous versions to address limitations and enhance performance [15].

Using the OpenCV and MediaPipe, a total of 501 landmarks (skeleton joints) were identified. These landmarks were then fed to a customized multiclass classification model to understand the relationship between each class and its coordinates for classifying and detecting a body posture [16]. The OpenCV library was first used for image processing. After that, the MediaPipe library was applied to extract x, y, and z coordinates, as well as the number of pixels for each joint. Finally, the YOLOv8 model was employed to identify the black cardboard, calculate the number of pixels in the cardboard, and determine the ratio of cm/pixel according to Equation (1):(1)$$(k)=\frac{height(cm)}{dis(pixel)}$$

The human body was divided into six segments: $h_{1}$ is the distance from the shoulder to the hip, $h_{2}$ is the distance from the hip to the knee, $h_{3}$ is the distance from the knee to the ankle, $h_{4}$ is the distance from the ankle to the sole of the foot, $h_{5}$ is the distance from the middle of the shoulder to the middle of the mouth, and $h_{6}$ is the distance from the middle of the mouth to the nose. The distance between two points, A($x_{a}$, $y_{a}$) and B($x_{b}$,$y_{b}$), was calculated. In this project, our calculations were based on the normalized coordinates $x_{i}$ obtained from MediaPipe $y_{i}$. These coordinates were then converted to a pixel coordinate system using Equations (2) and (3).
(2)$$X_{i}=image\_width*x_{i}$$
(3)$$Y_{i}=image\_height*y_{i}$$

The pixel coordinates were then used to calculate the distances between landmarks and the lengths of the skeleton segments in the human body. The coordinates of the midpoint of the shoulder and the coordinates of the midpoint of the hip were used to calculate the distance $h_{1}$ (Equation (4)):(4)$$h_{1}=k\times \sqrt{{\left(\frac{X_{23}+X_{24}}{2}-\frac{X_{11}+X_{12}}{2}\right)}^{2}+{\left(\frac{Y_{23}+Y_{24}}{2}-\frac{Y_{11}+Y_{12}}{2}\right)}^{2}}$$

The skeletal segment $h_{2}$ was calculated as the distance between points 23 and 25 (Equation (5)):(5)$$h_{2}=k\times \sqrt{{\left(X_{25}-X_{23}\right)}^{2}+{\left(Y_{25}-Y_{23}\right)}^{2}}$$

Similarly, the distance $h_{3}$ was calculated as the distance between points 27 and 25 (Equation (6)):(6)$$h_{3}=k\times \sqrt{{\left(X_{27}-X_{25}\right)}^{2}+{\left(Y_{27}-Y_{25}\right)}^{2}}$$

The distance $h_{4}$ from the ankle to the sole of the left foot was calculated as follows:(7)$$h_{4}=k\times \frac{\left|{(Y}_{29}-Y_{31})x_{27}+(X_{31}-X_{29})Y_{27}+{(Y}_{31}-Y_{29})X_{29}-(X_{31}-X_{29})Y_{29}\right|}{\sqrt{{\left(Y_{29}-Y_{31}\right)}^{2}+{\left(X_{31}-X_{29}\right)}^{2}}}$$

The distance $h_{5}$ was calculated as the distance from the midpoint of the shoulder to the midpoint of the mouth (Equation (8)):(8)$$h_{5}=k\times \sqrt{{\left((\frac{X_{11}+X_{12}}{2}-\frac{X_{9}+X_{10}}{2}\right)}^{2}+{\left(\frac{Y_{11}+Y_{12}}{2}-\frac{Y_{9}+Y_{10}}{2}\right)}^{2}}$$

Finally, the distance $h_{6}$ from the midpoint of the mouth to the nose was calculated as follows:(9)$$h_{6}=k\times \sqrt{{\left((X_{0}-\frac{X_{9}+X_{10}}{2}\right)}^{2}+{\left(Y_{0}-\frac{Y_{9}+Y_{10}}{2}\right)}^{2}}$$

### 2.2. Predicting Result of Height Measurement

A multivariable linear regression, which is an extension of a single-variable linear regression algorithm, was used to train and predict body height. This algorithm has proven to be highly effective in predicting outcomes based on two or more independent variables.

The multivariate linear regression [17] equation takes the following form:
Y = β_0_ + β_1_ × X_1_ + β_2_ × X_2_ + … + β_n_ × X_n_ + ε (10)
where Y is the dependent variable that needs to be predicted. X_1_, X_2_, …, X_n_ are the independent variables, and β_0_, β_1_, β_2_, …, β_n_ are the relationship coefficients.

After calculating the length of each skeletal segment, we applied a multivariate linear regression equation to predict human height. Equation (10) becomes the following:(11)$$h=\beta_{0}+\beta_{1}h_{1}+\beta_{2}h_{2}+\beta_{3}h_{3}+\beta_{4}h_{4}+\beta_{5}h_{5}+\beta_{6}h_{6}+\varepsilon$$
where h is the predicted height; h_1_, h_2_, h_3_, …, h_6_ are the calculated distance of skeleton segments; and β_0_, β_1_, β_2_, …, β_6_ are the correlation coefficients obtained during the process of training the multivariate linear regression model.

The multivariate linear regression model is an important tool for investigating relationships between several response variables and multiple predictor variables. The primary focus is on making inferences about the unknown regression coefficient matrices. We propose multivariate bootstrap techniques as a means for drawing inferences about these matrices. A real data example and two simulated data examples that provide finite sample verifications of our theoretical results are presented in [18,19].

### 2.3. Data Collection

This study was approved by the Hanoi University of Science and Technology. Data were collected from 166 adult subjects who agreed to participate in the study. Photographs of the subjects were taken with a smartphone camera. The smartphone was fixed on a tripod at a height of approximately 115 cm from the ground (Figure 2). The tripod was positioned at distances of 200 cm and 300 cm from the subject. A 20.5 cm × 30.5 cm black cardboard was placed next to the subject on the wall, with the center of the cardboard at a height of approximately 115 cm from the ground. On the opposite side of the subject, a wall height chart was attached.

Subjects were guided to perform four different postures during the experiment. Firstly, there was the standing-upright position (Figure 3a), where subjects stood straight and looked directly ahead. This position simulates the body in a natural state, with no tilt or rotation. Secondly, in the 45-degree rotation position (Figure 3b), subjects turned their bodies 45 degrees away from the camera while looking straight ahead. This pose simulates the body at a slight angle, which can affect how bone segments appear in the image. Thirdly, in the horizontal 90-degree rotation position (Figure 3c), subjects turned their bodies 90 degrees from the camera and looked straight ahead. This position simulates the body at a greater angle and illustrates patients’ positions in a hospital bed. This helps to better understand the differences in measurements of bone segments when the body is in a horizontal state, which is important in medical applications. Finally, in the kneeling position (Figure 3d), subject turned their bodies 90 degrees but bent their knees. This position is especially important for understanding changes in bone segments when the body is in a bent-knee state, simulating situations where the body is not completely upright. In each pose, subjects remained in position throughout the image capturing process to ensure the accuracy of the measurements. Staying steady and immobile during each scan is crucial to ensure that body landmarks are accurately and consistently identified.

### 2.4. Data Processing

The obtained images were processed using MediaPipe and YOLOv8 to extract the X and Y coordinates of the landmarks on the body, as well as the parameters of the reference object, which served for calculating the lengths of the skeletal segments. After that, the mean and standard deviation (SD) [19] were used to remove outliers to increase model accuracy. Specifically, data outside of the ±3 SD range were removed. The remaining valid values were used as input for the training model. Finally, a multivariate linear regression model was applied to the skeletal segments to estimate subjects’ height. The collected data consisted of 166 samples for each posture, with heights ranging from 148 cm to 184 cm. After eliminating outliners, a new dataset consisting of 162 samples was divided into 80% for training and 20% for testing.

## 3. Results

### 3.1. Standing Upright Position

After training the model with the training and test samples, we developed Equation (12) to estimate height based on bone segment lengths. Table 1 provides the evaluation results for the standing-upright position. This method has an average error of 1.94 cm (1.14%) across the test data samples. The error is mainly due to a lack of camera calibration and inaccuracies in extracting coordinates from MediaPipe, as well as varying lighting conditions during data collection.
(12)$$H=1.07533865h_{1}\;+1.2476316h_{2}\;+0.59605108h_{3}+0.6496244h_{4}+0.76927537h_{5}-2.13930107h_{6}+58.4779461$$

### 3.2. 45-Degree Rotation Position

Similarly, Equation (13) was developed to estimate body height for the 45-degree rotation position. Evaluation results for this position are presented in Table 2. The average error is 1.91 cm (1.12%).
(13)$$H=-0.70003005h_{1}+0.98866088h_{2}+0.76985497h_{3}+0.35090296h_{4}+0.68119476h_{5}-0.40682656h_{6}+72.229882$$

### 3.3. Horizontal 90-Degree Rotation Position

After training with the dataset for posture 3, Equation (14) was derived to estimate height for the 90-degree turned posture. Table 3 provides the evaluation results for the horizontal 90-degree rotation position. The average error is 2.62 cm (1.54%).
(14)$$H=0.08162038h_{1}\;+0.70345667h_{2}\;+0.67353882h_{3}\;+0.55258515h_{4}\;+0.42507086h_{5}-0.20956018h_{6}+73.384567$$

### 3.4. Kneeling Position

For the 90-degree sideways bent-knee position, we derived Equation (15) to calculate body height. Results are presented in Table 4. The evaluation results for this position show an average error of 2.45 cm (1.43%).
(15)$$H=0.36422493h_{1}+0.81095132h_{2}+0.58705451h_{3}+0.58410078h_{4}+0.35394516h_{5}-0.21066217h_{6}+73.779571$$

## 4. Discussion

The experimental results demonstrate that the proposed height estimation method can estimate human height relatively accurately, with an average error ranging from 1.91 cm to 2.62 cm (1.12–1.54%). Among the four postures, the height estimation model for the 45-degree rotation position yields the best results, with an average error of 1.91 cm (1.12%). For the other postures, the achieved results are less accurate. The standing-upright position has a result nearly equal to that of the 45-degree rotation posture, with an average error of 1.94 cm (1.14%). However, for the horizontal 90-degree rotation position and the kneeling position, the errors are significantly larger, with average errors of 2.62 cm (1.54%) and 2.45 cm (1.43%), respectively. This is mainly because the MediaPipe model does not perform as effectively when estimating height in more complex postures compared to the standing-upright posture. Additionally, other factors, such as lighting conditions and camera angles, also affect the accuracy of the measurement.

## 5. Conclusions

This study presents a non-contact height measurement method utilizing the MediaPipe library in conjunction with the YOLOv8 model to extract joint coordinates and calculate bone lengths, employing a multivariate linear regression function for predicting human height from images. Experimental results indicate that the average errors between the estimated and actual heights range from 1.91 cm to 2.62 cm (1.12% to 1.54%). This level of accuracy is deemed acceptable for a variety of applications. Future research will focus on expanding the methodology to determine the height of individuals in various standing and lying positions. The goal is to develop a flexible and efficient software application capable of measuring height across diverse real-world contexts. The integration of technologies such as MediaPipe and YOLOv8 demonstrates significant potential for applications in fields such as medicine, sports, and health monitoring, where reliable and precise height measurements from images are essential.

## Figures and Tables

**Figure 1 sensors-24-06796-f001:**
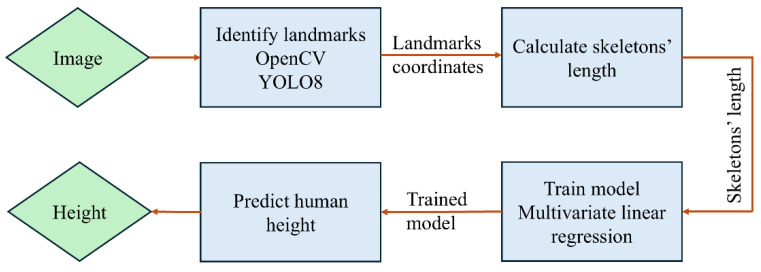
System diagram.

**Figure 2 sensors-24-06796-f002:**
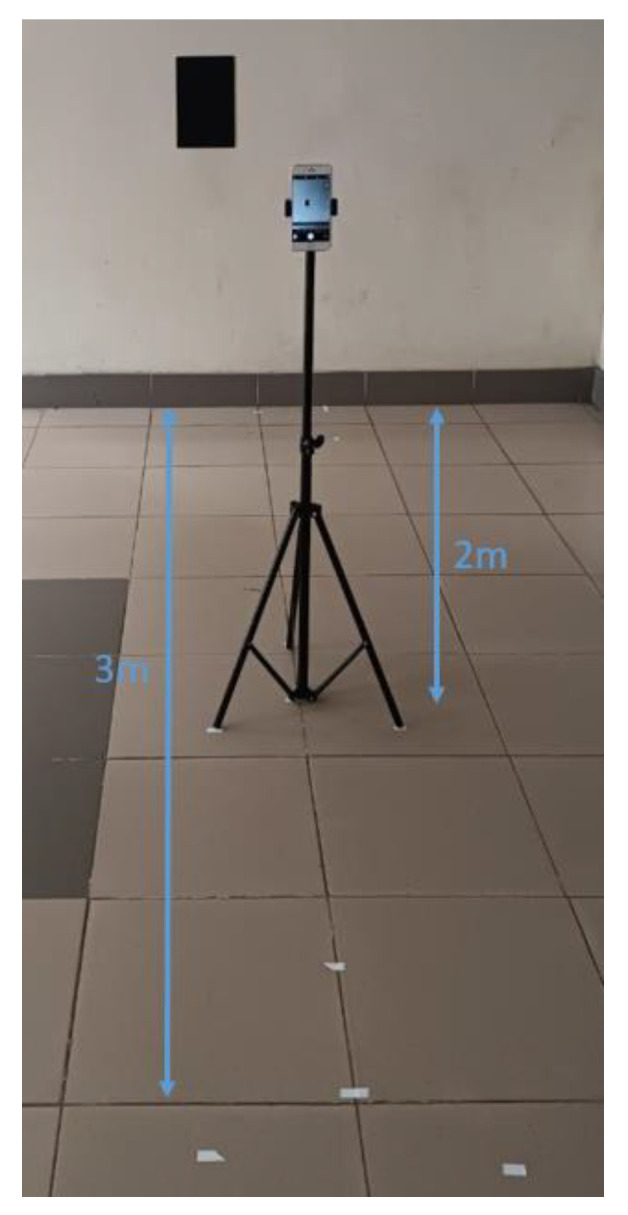
Tripod set up and camera.

**Figure 3 sensors-24-06796-f003:**
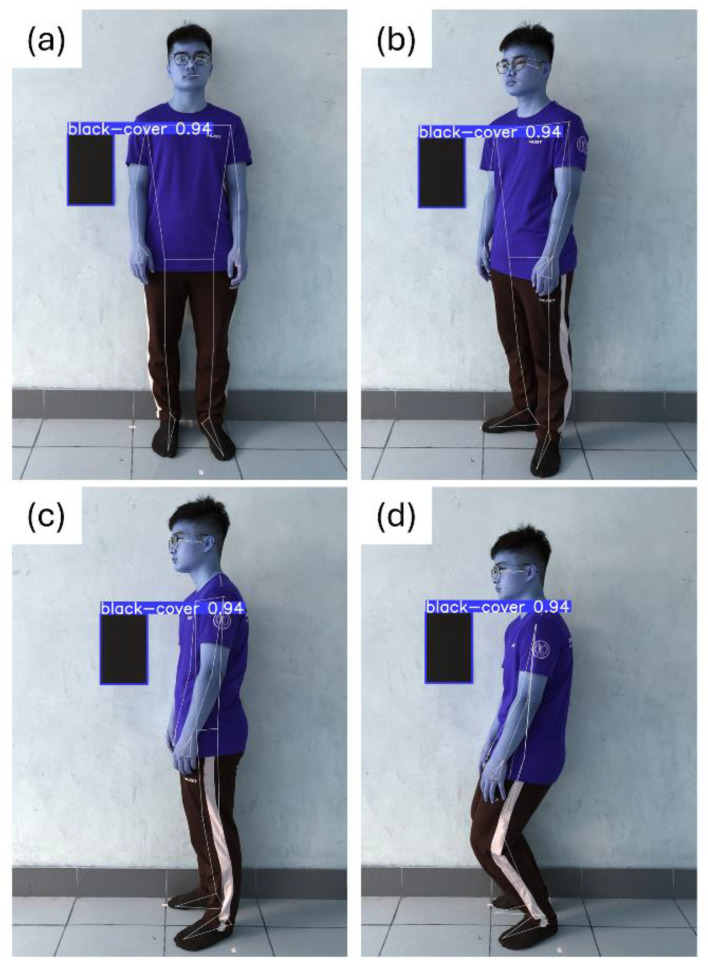
Height measurement in different postures; (**a**) standing-upright position; (**b**) 45-degree rotation position; (**c**) horizontal 90-degree rotation position; and (**d**) Kneeling position. Lines and points in each figure represent segments and joints determined from the OpenCV and the MediaPipe libraries.

**Table 1 sensors-24-06796-t001:** Prediction results for 17 subjects in the standing-upright posture.

Samples	Actual Height (cm)	Predicted Height (cm)	Error (cm)	Error Rate (%)
1	174	170.9514	3.048575	1.752055
2	177	177.4226	0.422636	0.238778
3	162	167.6994	5.699382	3.518137
4	168	168.3282	0.328223	0.195371
5	172	169.1883	2.811732	1.634728
6	169	172.8558	3.855773	2.281522
7	170	169.2551	0.744909	0.438182
8	170	173.3402	3.340176	1.964809
9	180	180.4499	0.449862	0.249923
10	175	173.0343	1.965713	1.123264
11	169	171.7793	2.779289	1.64455
12	175	175.5055	0.505452	0.28883
13	173	175.8168	2.816793	1.628204
14	168	170.0952	2.095163	1.247121
15	171	170.7831	0.216859	0.126818
16	168	167.1057	0.894323	0.532335
17	163	163.9993	0.999287	0.613059
Average	170.8	171.6241	1.939656	1.145746

**Table 2 sensors-24-06796-t002:** Prediction results for 17 subjects in the 45-degree tilted-standing posture.

Samples	Actual Height (cm)	Predicted Height (cm)	Error (cm)	Error Rate (%)
1	174	170.3426	3.657426	2.101969
2	177	178.7488	1.748834	0.988042
3	162	167.0036	5.003625	3.088657
4	168	167.3131	0.686861	0.408846
5	172	174.8213	2.821337	1.640312
6	169	166.2978	2.702194	1.598931
7	170	167.9283	2.071712	1.218654
8	170	168.6685	1.331542	0.78326
9	180	178.4254	1.574556	0.874753
10	175	174.1332	0.866795	0.495312
11	169	171.0258	2.025811	1.198705
12	175	175.7314	0.731434	0.417962
13	173	174.0368	1.03684	0.599329
14	168	164.0762	3.923809	2.335601
15	171	172.5887	1.588712	0.929071
16	168	167.4733	0.526727	0.313528
17	163	162.7955	0.20453	0.125479
Average	170.8	170.6712	1.911926	1.124612

**Table 3 sensors-24-06796-t003:** Prediction results for 17 subjects in the 90-degree tilted-standing posture.

Samples	Actual Height (cm)	Predicted Height (cm)	Error (cm)	Error Rate (%)
1	177	177.8865	0.886495	0.500845
2	162	169.2808	7.2808	4.494321
3	168	168.0163	0.016319	0.009714
4	172	169.4318	2.568204	1.493142
5	169	168.173	0.826959	0.489325
6	170	176.0739	6.073928	3.572899
7	170	168.528	1.47199	0.865877
8	180	183.2167	3.216669	1.787039
9	175	171.7336	3.266446	1.86654
10	169	174.7692	5.769226	3.413743
11	169	168.7802	0.219849	0.130088
12	175	172.8844	2.115569	1.208896
13	173	175.1489	2.148942	1.242163
14	168	165.562	2.438041	1.451215
15	171	173.9013	2.901262	1.696644
16	168	167.8982	0.10178	0.060583
17	163	166.2274	3.22743	1.980018
Average	170.5	171.6184	2.619406	1.544885

**Table 4 sensors-24-06796-t004:** Prediction results for 17 subjects in the 90-degree tilted-standing posture with bent knees.

Samples	Actual Height (cm)	Predicted Height (cm)	Error (cm)	Error Rate (%)
1	174	169.3661	4.633906	2.663164
2	177	178.5004	1.50038	0.847672
3	162	163.4223	1.422282	0.877952
4	168	168.3369	0.336886	0.200527
5	172	170.5549	1.445134	0.840194
6	169	168.7444	0.255635	0.151263
7	170	176.0042	6.004188	3.531876
8	170	175.1093	5.109334	3.005491
9	180	182.6475	2.647498	1.470832
10	175	178.3741	3.374108	1.928062
11	169	172.6141	3.614117	2.138531
12	175	172.9836	2.01644	1.152252
13	173	175.8976	2.897555	1.674887
14	168	166.5528	1.44725	0.861458
15	171	169.4226	1.577427	0.922472
16	168	167.0025	0.997496	0.593748
17	163	165.3323	2.332332	1.430879
Average	170.8	171.8156	2.447763	1.428898

## Data Availability

Data will be available from the corresponding authors upon reasonable request.

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
