# Peer review of "A Non-Contacted Height Measurement Method in Two-Dimensional Space"

_sensors, 2024, doi:10.3390/s24216796_

Round 1

Reviewer 1 Report

Comments and Suggestions for Authors

The manuscript presents a new method to measure human height non-contactly from various poses and distances using a smartphone camera, which includes data collection, image processing, and height calculation methods. The method has been evaluated through a series of experiments demonstrating accurate height measurement results.

There are some minor limitations which should be corrected, including:

- Equation in the same line with text: equations 1, 2, 4, and 5 should be presented on a separate line to be consistent with other equations. 

- The discussion section is quite short. It should discuss the summary of the results in all four positions. It also could be more specific on which shooting angle may negatively affect the measurement results. 

- The errors should be reported in both `cm' and % in each result discussion and the conclusion. Currently, the `cm' error is only present in tables but not in the writing. Many related studies and applications of human height measurement are interested in the actual error range, not the percentage.

- Some places mention 'potion 1' or 'pose 1', position 2, position 3. However, they should be specific, such as standing upright, especially in the paper's discussion section. If the author would like to keep the number identification of the position, they should identify these position numbers at each posture description presented in Section 2.3, page 5, lines 174 to 187. 

Comments on the Quality of English Language

Minor grammar mistakes, such as:

Page 9, line 271: not require a comma:  "software application, capable of" 

Page 9, line 273: need a comma: "health monitoring and other fields"

Problematic sentences, such "The results are presented in Table 1 Evaluation results for position 1." This mistake happened in sections 3.1, 3.2, and 3.3. It should not use the capital word 'Evaluation' in the middle of the sentence.

Suggestions of change are:

"Evaluation results for the standing upright position are presented in Table 1." 

Or "Table 1 provides the evaluation results for the standing upright position."

Author Response

Reviewer 1:

The manuscript presents a new method to measure human height non-contactly from various poses and distances using a smartphone camera, which includes data collection, image processing, and height calculation methods. The method has been evaluated through a series of experiments demonstrating accurate height measurement results.

Response: We sincerely thank the reviewer for your valuable comments and suggestions. We have changed our manuscript and highlighted the changes in yellows. Below please find our responses (in blue) to your questions and comments.

There are some minor limitations which should be corrected, including:

- Equation in the same line with text: equations 1, 2, 4, and 5 should be presented on a separate line to be consistent with other equations. 

Response: Equations 1, 2, 3, 4, and 5 have been adjusted to separate lines on page 3 at lines 103 (equation 1), 111 (equation 2), 115 (equation 3), 122 (equation 4), and 125 (equation 5).

- The discussion section is quite short. It should discuss the summary of the results in all four positions. It also could be more specific on which shooting angle may negatively affect the measurement results. 

Response: The discussion section has been rewritten while maintaining the main content as before. Additional discussion has been added summarizing the results for all four positions.

- The errors should be reported in both `cm' and % in each result discussion and the conclusion. Currently, the `cm' error is only present in tables but not in the writing. Many related studies and applications of human height measurement are interested in the actual error range, not the percentage.

Response: Error reporting in cm has been added to the conclusion and the results for each posture.

- Some places mention 'potion 1' or 'pose 1', position 2, position 3. However, they should be specific, such as standing upright, especially in the paper's discussion section. If the author would like to keep the number identification of the position, they should identify these position numbers at each posture description presented in Section 2.3, page 5, lines 174 to 187. 

Response: The titles of sections 3.1, 3.2, 3.3, and 3.4 have been changed to synchronize with the postures described in section 2.3.

"The 45-degree turned pose" has been replaced with "the 45-degree rotation position" 

Comments on the Quality of English Language

Minor grammar mistakes, such as:

Page 9, line 271: not require a comma:  "software application, capable of" 

Response: The comma has been removed from the phrase "software application, capable of"

Page 9, line 273: need a comma: "health monitoring and other fields"

Response: A comma has been added to make it "health monitoring, and other fields"

Problematic sentences, such "The results are presented in Table 1 Evaluation results for position 1." This mistake happened in sections 3.1, 3.2, and 3.3. It should not use the capital word 'Evaluation' in the middle of the sentence.

Suggestions of change are:

"Evaluation results for the standing upright position are presented in Table 1." 

Or "Table 1 provides the evaluation results for the standing upright position."

Response: The sentence "The results are presented in Table 1 Evaluation results for position 1." in section 3.1 has been changed to "Table 1 provides the evaluation results for the standing upright position."

The sentence "The results are presented in Table 3 Evaluation results for position 2." in section 3.2 has been changed to "Evaluation results for the 45-degree rotation position are presented in Table 3."

The sentence "The results are presented in Table 5 Evaluation results for position 3." in section 3.3 has been changed to "Table 5 provides the evaluation results for the horizontal rotation position 90 degrees."

Reviewer 2 Report

Comments and Suggestions for Authors

This paper needs significant revision to improve its motivation, review of existing related works, methodology presentation, equations, figures, and whole paper organization. The current version is in the form of a coarse technical report.

Comments on the Quality of English Language

The English usage is ok but can also be improved, e.g., shorten the excessively long sentences be make them more concise.

Author Response

This paper needs significant revision to improve its motivation, review of existing related works, methodology presentation, equations, figures, and whole paper organization. The current version is in the form of a coarse technical report.

Response: Thank you so much for your valuable comments. The manuscript has been revised significantly. Please find revision highlighted in yellow in the manuscript.

Comments on the Quality of English Language

The English usage is ok but can also be improved, e.g., shorten the excessively long sentences be make them more concise.

Response: Excessively long sentences have been shortened.

Reviewer 3 Report

Comments and Suggestions for Authors

The authors develop a non-contact method for height measurement of human. Overall, the method is easy-to-follow but lack of systematically study literature and novelty. The authors must thoroughly enrich literature research and algorithm complexity.

1."measure human height from 2 to 3 meters" in abstract is unrealistic.

2. Innovation is mainly reflected in non-contact. Expand description based on contact and non-contact and briefly introduce the motivation behind this innovation

3. Beautify Figure 1.

4. Reduce the naivety of algorithms and omit basic introduction, such as distance formula for vectors.

5. Merge Table 1 and 2, the rest is the same. Add other indicators for evaluating algorithms, such as CPU running time and standard deviation.

Comments on the Quality of English Language

Moderate editing of English language required

Author Response

The authors develop a non-contact method for height measurement of human. Overall, the method is easy-to-follow but lack of systematically study literature and novelty. The authors must thoroughly enrich literature research and algorithm complexity.

Response: We would like to send our special thanks to the reviewer for your dedicated time to review our manuscript and provide us with valuable feedback. We have applied your comments and suggestions to improve our manuscript. The changes in our manuscript have been highlighted in yellow for your convenience. Below please find our responses (in blue) to your feedback.

1."measure human height from 2 to 3 meters" in abstract is unrealistic.

Response: In the phrase "measure human height from 2 to 3 meters”, the authors mean to state that human height can be measured from a distance of 2 to 3 meters. In order to avoid the confusion, the phrase "from a fixed distance" in the abstract has been removed.

  1. Innovation is mainly reflected in non-contact. Expand description based on contact and non-contact and briefly introduce the motivation behind this innovation

Response: Information about contact method and motivation of this study has been added to the introduction.

  1. Beautified Figure 1.

Response: Figure 1 has been beautified.

  1. Reduce the naivety of algorithms and omit basic introduction, such as distance formula for vectors.

Response: Basic introduction has been removed.

  1. Merge Table 1 and 2, the rest is the same. Add other indicators for evaluating algorithms, such as CPU running time and standard deviation.

Response: Table 1 and 2 have been merged. Similarly, table 3 & 4, 5&6, and 7&8 have been merged.

Comments on the Quality of English Language

Moderate editing of English language required

Response: English language has been modified throughout the manuscript.

Round 2

Reviewer 3 Report

Comments and Suggestions for Authors

The authors have been modified the manuscript according to my advise, and thus I recommend acceptance.

Comments on the Quality of English Language

Moderate editing of English language required.

Author Response

We sincerely thank the reviewer for your comment. We have carefully revised the manuscript for the English language. Please find our change in the manuscript, which has been highlighted in yellow.